# Bone Disorders in Pediatric Chronic Kidney Disease: A Literature Review

**DOI:** 10.3390/biology12111395

**Published:** 2023-11-02

**Authors:** Lavinia Capossela, Serena Ferretti, Silvia D’Alonzo, Lorenzo Di Sarno, Valeria Pansini, Antonietta Curatola, Antonio Chiaretti, Antonio Gatto

**Affiliations:** 1Institute of Pediatrics, Fondazione Policlinico A. Gemelli IRCCS, Università Cattolica Sacro Cuore, 00168 Rome, Italy; serena.ferretti01@icatt.it (S.F.); lorenzodisarno1993@gmail.com (L.D.S.); antonio.chiaretti@policlinicogemelli.it (A.C.); 2Nephrology Unit, Department of Medical and Surgical Sciences, Fondazione Policlinico Universitario A. Gemelli IRCSS, 00168 Rome, Italy; silvia.dalonzo@policlinicogemelli.it; 3Institute of Pediatrics, Fondazione Policlinico Universitario A. Gemelli IRCCS, 00168 Rome, Italy; pansini.valeria@gmail.com (V.P.); c.anto91@hotmail.it (A.C.); antonio.gatto@policlinicogemelli.it (A.G.)

**Keywords:** bone, disease, renal, children, osteodystrophy

## Abstract

**Simple Summary:**

Mineral and bone disorder (MBD) is usually prevalent in pediatric patients with chronic kidney disease (CKD) and is correlated with meaningful morbidity. CKD may cause disorders in bone remodeling/modeling, which are more evident in the growing skeleton, expressing as short stature, bone pain and deformities, fractures, slipped epiphyses and ectopic calcifications. Although evaluation of bone health is a crucial part in the clinical care of children with CKD, it persists as a significant challenge for pediatricians.

**Abstract:**

Intense changes in mineral and bone metabolism are frequent in chronic kidney disease (CKD) and represent an important cause of morbidity and reduced quality of life. These disorders have conventionally been defined as renal osteodystrophy and classified based on bone biopsy, but due to a lack of bone biopsy data and validated radiological methods to evaluate bone morphology in children, it has been challenging to effectively assess renal osteodystrophy in pediatric CKD; the consequence has been the suboptimal management of bone disorders in children. CKD–mineral and bone disorder (CKD-MBD) is a new expression used to describe a systemic disorder of mineral and bone metabolism as a result of CKD. CKD-MBD is a triad of biochemical imbalances in calcium, phosphate, parathyroid hormone, and vitamin D; bone deformities and soft tissue calcification. This literature review aims to explore the pathogenesis, diagnostic approach, and treatment of CKD-MBD in children and the effects of renal osteodystrophy on growing skeleton, with a specific focus on the biological basis of this peculiar condition.

## 1. Introduction

Chronic renal failure is defined by intense changes in the ordered metabolic sequences, which usually assurance cellular integrity and metabolic homeostasis [1].

The kidney plays a significant role in the regulation of calcium, inorganic phosphate, parathyroid hormone, calcitonin, and vitamin D metabolism. Adults and children with progressive loss of renal parenchyma suffer from modifications in bone metabolism with resultant osteodystrophy [2]. Renal osteodystrophy is the term used to delineate the bone morphology related with chronic kidney disease (CKD) [3]. It represents a variety of skeletal lesions that range from high-turnover conditions (osteitis fibrosa and mild lesions of secondary hyperparathyroidism) to low-turnover bone diseases (osteomalacia and adynamic lesions) [4].

In 2006, the KDIGO (Kidney Disease Improving Global Outcomes) introduced the inclusionary term of chronic kidney disease–mineral and bone disorder (CKD-MBD) to describe this clinical entity [5]. MBD is a triad of biochemical abnormalities (calcium, phosphate, parathyroid hormone (PTH) and 1,25-dihydroxyvitamin D), bone abnormalities (short stature, reduced mineralization, and increased risk of fractures), and extra-skeletal calcification [6].

Complications of CKD-MBD include vascular calcification, stroke, skeletal fractures, and increased risk of death. Increased FGF23 and PTH concentrations and 1,25 dihydroxy vitamin D (1.25(OH)2D) deficiency support the pathogenesis of CKD-MBD [7].

Due to an absence of bone biopsy data and validated radiologic methods to evaluate morphology of the bone, it has been difficult to adequately assess renal osteodystrophy in pediatric CKD. This has directed suboptimal management of bone disorders in children.

This literature review aims to explore the pathogenesis, diagnostic approach, and treatment of CKD-MBD in children and the effects of renal osteodystrophy on growing skeleton, with a specific focus on the biological basis of this peculiar condition.

## 2. Methods and Results

We conducted a literature review of the past 15 years to find renal osteodystrophy-related studies and reports. The following electronic databases were systematically searched: PubMed, Scopus, and Cochrane Central Register of Controlled Trials (CENTRAL).

The research strings were:Renal AND osteodystrophy;Renal AND bone;Bone mineral AND renal;Bone disorder AND renal;CKD-MBD AND children.

Papers were only included in the present review if they focused on the pediatric population (ranging from 0 months to 18 years old). Only English-written publications were included. The current investigation mainly concentrated on randomized placebo-controlled studies, but case–control studies, retrospective and prospective observational studies, and systematic reviews and meta-analysis were evaluated as well.

The article selection method was supported independently by three reviewers (LC, SF, and LDS).

All significant articles discovered were further scrutinized for extra references not appearing in the preliminary examination.

Review or commentary papers without original data were eliminated, whereas their contents were used for clarification of collected information.

All papers not noticeably concerning renal osteodystrophy or CKD-MBD or regarding only adult CKD-MBD were similarly excluded.

The quality of the trials was thoroughly evaluated and the following potential biases have been assessed: random classification group (selection bias), similarity of patients at baseline concerning the most significant prognostic indicators (homogeneity bias), allocation hiding (selection bias), blinding of workers (performance bias), blinding of outcome valuation (detection bias), partial outcome data (attrition bias), avoidance of co-interventions (co-intervention bias), and reports of drop-outs (drop-out bias).

A total of 63 papers were encountered in the literature search. Among them, after the application of inclusion and exclusion criteria, 55 papers were selected, while 8 papers were excluded.

### 2.1. Pathogenesis

The pathogenesis and natural history of CKD-MBD is mainly connected to secondary hyperparathyroidism [8]. Renal failure causes hypocalcemia and hyperphosphatemia. The former is due to reduced 1-alpha-hydroxylation of 25-hydroxyvitamin D, which leads to low intestinal absorption of calcium. Hypocalcemia stimulates parathyroid glands to produce more PTH. On the other hand, reduced glomerular filtration rate (GFR) causes phosphate retention, which stimulates the osteocytes to secrete FGF23. These pathways lead to increased phosphaturia by PTH action on the renal phosphate sodium co-transporter and mobilize calcium out of bone, affecting its mineralization [9]. Moreover, FGF23 inhibits WNT pathways, which contributes to bone degradation and consequent higher fracture risk [10].

In the last decade, new mechanisms and regulatory molecules have been identified, including alpha-Klotho, FGF-23 and its receptor, vitamin D receptors, sclerostin, Ca-sensitive receptors, and decarboxylated osteocalcin, which have allowed us to shed greater light on the pathogenesis of CKD-MBD. While the “traditional” factors seem to become relevant only in the advanced stages of CKD, the new factors would already act in the early stages of kidney disease [11,12].

It is now believed that the first alteration in the initial stages of CKD-MBD is an increase in circulating levels of FGF-23; therefore, the dosage of this factor can represent an important marker for early identification of alterations in mineral and bone metabolism. It is not fully understood what the first stimulus that induces an increase in FGF-23 levels is. In the first instance, it is believed that it follows (or is simultaneous with) a reduction in the production of alpha-Klotho, a transmembrane protein that regulates the activation of FGF-23 receptors at the level of the target organs (kidney and parathyroid) and osteocytes, and whose levels modulate those of growth factor through a negative feedback mechanism. In the later stages of CKD, a progressive reduction in renal function is accompanied by a strong increase in FGF-23 levels, together with hyperphosphatemia, calcitriol deficiency, and hypocalcemia. These mechanisms result in persistently elevated PTH values, usually seen when GFR values decrease below 50 mL/min/1.73 m^2^ [8].

Blood calcium levels decrease not only as a consequence of calcitriol deficiency, but also due to phosphate (P) retention and the development of bone resistance to PTH action. Hypocalcemia represents an important stimulus to PTH secretion, mediated by calcium sensitive receptors (CaSR), which are highly expressed in parathyroid cells. However, over the long term, vitamin D receptors and CaSRs develop resistance to the stimulus, which can lead to hyperplasia of the parathyroid glands (tertiary hyperparathyroidism) in some patients [8].

Finally, although in the early stages elevated FGF-23 levels inhibit PTH secretion, keeping PTH levels low, it is believed that, in advanced stages of CKD, parathyroid cells become resistant to this stimulus.

These pathogenetic mechanisms lead to an alteration in the turnover, mineralization, and bone volume, with consequent alterations in the development of bone histomorphometry, responsible for poor growth in childhood and for osteomalacia to fibrous osteitis in adulthood. In particular, hyperparathyroidism increases bone turnover, mainly by inducing the expression of RANKL (receptor activator of nuclear factor kappa B ligand) on the surface of osteoblasts, with consequent differentiation of the latter into osteoclasts. The simultaneous lack of calcitriol, causing a serious lack of bone mineralization, leads to the growth of bone with a reduced mineral mass and therefore more exposed to the risk of fractures [13].

Furthermore, it is recent knowledge that the maturation of the skeleton is accompanied by the development of muscle mass. In particular, it is believed that the latter anticipates bone development, representing a crucial element for bone health during childhood and adolescence (musculoskeletal axis) [14].

Patients with CKD show muscle atrophy and weakness in 7–20% of cases, especially in the advanced stages of the disease [8]. The resulting osteo-sarcopenia exposes them to a greater risk of fractures, disability, and hospitalization [14]. Again, vitamin D deficiency and elevated PTH levels have a negative effect on muscle. The former alters contractile force and metabolism of muscle by activating the expression of myogenic transcription factors and contractile proteins. The latter, however, stimulates muscle proteolysis and alters energy metabolism. The effect is a marked reduction in muscle strength, currently well documented in children with CKD [8].

Furthermore, the inflammatory process increases muscle proteolysis and promotes the expression of myostatin (a factor limiting muscle growth). Anorexia, frequent in these subjects, contributes to the depletion of muscle mass [8,15].

### 2.2. New Classification

Biochemical anomalies in the serum levels of phosphorus, calcium, PTH, and vitamin D lead to multiple bone alterations in patients with CKD. Both the laboratory modifications and the bone deformities contribute to vascular calcification [16]. All three of these processes are strictly connected and justify the significant morbidity and mortality in patients with CKD.

Traditionally, renal osteodystrophy has always been classified only according to a small subset of indicators associated with mineral and bone disorders in CKD as stated above. On the contrary, the modern definition should include all the following three main features: serum biomarkers, noninvasive imaging (in order to assess extra-skeletal calcification), and bone anomalies.

The KDIGO (Kidney Disease: Improving Global Outcomes) sponsored a summit in 2005 on the reconsideration of the current descriptive terminology for this pathophysiologic process [5]. The conference suggested that the term renal osteodystrophy be used solely to describe the histologically documented abnormal bone disease associated with CKD [17]. In order to diagnose renal osteodystrophy, the gold standard is represented by a bone biopsy, which also allows to properly classify this entity. Interestingly, the summit also stated that renal osteodystrophy should only be considered to be one of the many elements that can be referred to by the new clinical entity or syndrome named chronic kidney disease–mineral and bone disorder (CKD-MBD). In fact, the conference agreed that the definition of CKD-MBD should include elements of abnormal mineral metabolism, altered bone configuration and structure, and extra-skeletal calcification documented with clinical, biochemical, and imaging findings.

The early KDIGO guideline on CKD-MBD was then published in 2009 [17]. New evidence was then revised at the 2013 KDIGO Controversies Conference, and in 2017, KDIGO released a clinical practice guideline update for the diagnosis, evaluation, prevention, and treatment of CKD-MBD [18,19,20].

### 2.3. Abnormalities of Bone in CKD

Bone remodeling is a physiologic process in both adults and children and disorders of mineral metabolism are related to abnormal bone [16].

As for macrostructure, skeletal deformity is frequent in children with CKD-MBD and is typical of rapidly growing bones. Therefore, rickets radiographically presents with slipped epiphyses and other deformities such as bow legs and knock knees [9,21]. In addition, the impaired quality of the bone in CKD is a major cause of hip fractures in patients undergone dialysis [16,22].

On the other hand, inadequate vitamin D supplementation, imbalances of serum calcium and phosphate, and metabolic acidosis contribute to poor mineralization, with histologic findings of osteomalacia, typical of children affected by CKD-MBD [23].

Although bone biopsy is still considered the gold standard to diagnose CKD-MBD, its use is predominantly implicated solely in clinical research because of its invasive nature and the current incapability of pathology laboratories to interpret results [23].

In fact, before 1990, the most frequent skeletal lesion found at bone biopsy were increased bone formation and osteoclast and osteoblast activity and number, characteristic of secondary hyperparathyroidism [23,24]. However, the use of phosphate-binding agents and the iatrogenic reduction in PTH serum concentration have histologically led to adynamic bone, with low or normal bone formation rate and a reduced number of osteoclasts and osteoblasts [25]. Currently, different authors are studying the clinical implications of this new histologic finding, which are still not clear [23].

### 2.4. Assessing Bone Mineralization

In children with CKD, it is necessary to assess the clinical history and conduct a physical examination to find CKD-MBD-related bone disease. The frequency of evaluation is based on the principal cause and stage of CKD, the age of the patient, the symptoms, the existence of comorbidities, and the level of abnormalities in previous CKD-MBD measures. More numerous assessment during periods of rapid growth in infancy and adolescence is needed (Table 1).

In children with CKD, further examinations to calculate linear growth rate are also needed; in particular, infants with CKD G2–G5D should have their length measured at least quarterly, while children with CKD G2–G5D should be evaluated for linear growth at least annually [16]

Biochemical markers show low sensitivity and specificity. As a marker of CKD-MBD, dosage of serum and ionized calcium, phosphate, alkaline phosphatase, PTH and 25(OH)D should be evaluated in children with CKD Stages 2–5D. Furthermore, it is necessary to frequently monitor serum bicarbonate levels and to maintain them within the normal range for the risk of metabolic acidosis. The severity of anomalies, the age, signs and symptoms, concomitant treatments, and the stage and progression of CKD should define the frequency of monitoring the disease. Age-related normal ranges of serum calcium, phosphate, alkaline phosphatase, and CKD stage-dependent PTH target ranges should be considered in the identification and management of bone disease in pediatric CKD.

Age-dependent normal values for different markers of bone and mineral metabolism and CKD stage-dependent PTH target ranges proposed by international guideline committees are based on the CALIPER study [1,20,26].

The 2017 KDIGO guidelines recommended examining serum levels of calcium, phosphate, PTH, and alkaline phosphatase activity at the beginning of CKD G3a. In children, they suggested this examination beginning in CKD G2. They stated to use trends in PTH rather than absolute “target” values to guide therapy as to start or stop treatments to lower PTH. If PTH trends are inconsistent, a bone biopsy may be contemplated.

Bone imaging and histology are variably used to assess bone disease in CKD, but there are limited evidence-based studies to encourage their use in routine clinical practice (Table 2).

Conventional X-rays can only roughly assess radiological findings, such as bone mineralization. Children affected by bone pain, doubted atraumatic fractures, and genetic diseases with a documented bone involvement should be assessed with this exam.

DXA, peripheral quantitative computed tomography (pQCT), high-resolution peripheral quantitative computed tomography (HR-pQCT), magnetic resonance imaging (MRI) and ultrasound are remarkable for research [7,27], but there is no evidence to recommend them as routine screening exams for bone health or evaluation of fracture risk in pediatric CKD patients.

The 2009 KDIGO CKD-MBD guideline noted that DXA BMD (bone mineral density) does not differentiate among categories of renal osteodystrophy. Furthermore, no findings have assessed the connection between DXA results and fractures, and so the KDIGO 2017 update does not offer specific recommendations for DXA use in children [20]

The advice to execute a bone biopsy is based on the prospect it may modify treatment and the choice to administer antiresorptive therapy. Since this is infrequently (if ever) performed in children with CKD, the role of bone biopsies remains debatable [3].

According to the literature, a bone biopsy is suggested in cases where laboratory and clinical results do not confirm an underlying osseous disease. Given the complex interpretation of the specimens, it should be considered to send the samples to experienced centers for a proper histomorphometric assessment. In fact, a proper diagnosis is mandatory in order to let the physician tailor a specific therapeutic strategy for the patient [1].

### 2.5. Cardiovascular Manifestations and Vascular Calcification

Cardiovascular disease is a major cause of mortality in CKD [28]. The intricate nature of renal, bone, and cardiovascular disease led to it being retitled as mineral and bone disorder of chronic kidney disease to emphasize how bone disease drives vascular calcification and confers the development of long-term cardiovascular complications. Recent data suggest that good management of the bone disease can increase and improve cardiovascular disease status [29].

Patients with CKD showed an increased risk of coronary artery disease due to calcium deposition and consequent arterial stiffening, in addition to left ventricular dysfunction with concomitant heart failure and arrhythmias. Children and young adults with CKD or on dialysis develop vascular calcification, even as bone mineral density rises, with the most significant vascular alterations occurring in young people with no linear growth [30].

While evidently affected by the traditional risk factors for development of cardiovascular disease, patients with CKD are also disturbed by non-traditional risk factors, including calcium overloading related to the forceful management of secondary hyperparathyroidism [31].

Recent data have shown that a considerable number of patients with CKD are lacking vitamin D on a nutritional basis, in addition to the known decrease in the kidney-produced active metabolite during progressive CKD. Vitamin D analogues are central in cardiovascular health. Pilot studies suggest that vitamin D therapy for secondary hyperparathyroidism may confer cardiac protection and decrease mortality [32]. Attention to osteodystrophy care in chronic kidney disease should also include heart health.

Furthermore, there is an amplified risk of obesity and metabolic syndrome among kidney transplant beneficiaries, which negatively affects cardiovascular and renal outcomes in these patients.

In fact, the increased prevalence of overweight children among the population affected by CKD configures the paradoxical of sarcopenic obesity, which is responsible for the increase in consumption of so-called empty-calorie foods, such as fast food and snacks. These data are worrying, as patients with CKD already present a high risk of cardiovascular disease, and excess weight can have a negative impact on the outcome of a kidney transplant [33].

Cardiometabolic risk factors are usual in pediatric kidney transplant recipients. Approximately one-fifth of patients have a metabolic syndrome, and left ventricular hypertrophy is much more common among patients with a metabolic syndrome [34,35,36].

### 2.6. Anemia in Pediatric CKD-MBD

Anemia is a common complication in children with CKD, which occurs in parallel with worsening kidney function, typically when the GFR falls below 43 mL/min/1.73 m^2^ [37]. It has long been known that a predominant role in the genesis of anemia is played by the reduced production of erythropoietin (EPO), but it has been recently found that its synthesis is stimulated by the factor inducible by hypoxia-1 (HIF-1), whose activity increases in cases of oxygen deficiency. In the course of CKD, the reduced consumption of oxygen by the renal tissue causes an increase in the partial pressure of the gas, resulting in a reduced stimulus to the synthesis of HIF1 and therefore EPO. Furthermore, circulating EPO binds to specific transmembrane receptors of erythroblasts, regulating their proliferation and maturation into erythrocytes. Following renal damage, the cells responsible for its production (interstitial cells of the peritubular capillaries) differentiate into myofibroblasts and acquire the ability to produce collagen, losing the ability to synthesize the hormone.

An important novelty in CKD is also the elevated levels of hepcidin, a protein that regulates the intestinal absorption of iron (Fe) and its distribution. This molecule binds to ferroportin expressed on cell membranes, causing their internalization and degradation, resulting in reduced Fe absorption at the enterocyte-level inhibition of its release by the cells of the reticuloendothelial system, such as to prevent its use for bone marrow erythropoiesis. In CKD, hepcidin levels are typically increased as a result of reduced EPO production and increased levels of circulating IL6 [38]. Also, the accumulation of uremic toxins and stress oxidative associated with it, potentially damaging the cell membrane and the cytoskeleton of erythrocytes, contribute to the onset of anemia, causing a reduction in the average lifespan of these cells. Finally, hyperparathyroidism itself contributes to the development of anemia and is above all a cause of resistance to therapy with recombinant EPO as an effect of the reduced production of red blood cells in the context of a fibrotic marrow [39].

Clinically, anemia has been associated with an increase in hospitalizations and mortality, linked to an increase in average blood pressure values and a reduction in school performance and the ability to carry out physical activities. Finally, there is the possibility that CKD patients may undergo blood transfusions, with consequent development of anti-HLA antibodies and increased risk of rejection in case of renal transplant. It is therefore important to monitor, from the early stages of CKD, hemoglobin levels and Fe status, together with folate and vitamin B12 levels, with variable frequency in relation to the stage of the disease [39].

The gold standard for the treatment of anemia associated with CKD is represented by the use of recombinant human erythropoietin (rHuEPO). KDIGO recommendations suggest starting therapy with rHuEPO in case of Hb values < 10 g/dl, with the aim of reaching a value between 11 and 12 g/dl. Such therapy must be undertaken only after correction of any iron and folate deficiency and after excluding the presence of other causes of anemia unrelated to CKD [40].

### 2.7. Diagnosis and Management of Mineral and Bone Disorders in Infants with CKD

Because of the rapid bone growth and difficulties in nutrition, an adequate control of CKD-MBD in infants is particularly challenging [41].

Infants with CKD are mainly susceptible to mineral and bone disorders. Infancy is the period of most rapid development with consequent elevated demands of calcium (Ca) and phosphate (P) to make a satisfactory mineral equilibrium and endochondral ossification [4].

CKD in infants exhibits persistent abnormal creatinine levels above the 97.5th percentile in the first 12 months of life and an estimated glomerular filtration rate (eGFR) < 90 mL/min per 1.73 m^2^ by the Schwartz formula [42,43].

The diagnosis of CKD-MBD in infants begins from physical examination for clinical signs of CKD-MBD, established by the infant’s corrected gestational age, stage of CKD, comorbidities, and severity of MBD. Systematic evaluation of development (weight, length, and head circumference), and serially schemed centile growth charts are necessary.

Concerning biochemical values, the measurement of CKD-MBD biomarkers (Ca, P, ALP, PTH, 25(OH)D, and HCO3) need to be regularly evaluated based on the infant’s (corrected) age, stage of CKD, underlying disease, and the presence and severity of MBD.

Management is not established on an individual laboratory value, but on all available CKD-MBD markers.

In this age group, it is essential to maintain serum Ca, P, alkaline phosphate, and HCO3 within the age-related normal ranges and PTH levels within the CKD-stage related target ranges. Moreover, the maintenance of 25(OH)D within the target range reported in older children is recommended.

Regarding radiological exams, the evidence indicates to not perform routine X-rays in infants with CKD, but plain X-rays should be prescribed in infants with clinical doubt of rickets or other bone disturbances. It should also be kept in mind that an adapted approach to radiological monitoring of infants with genetic diseases showing specific bone anomaly is important.

Regarding diet, calcium and phosphate intake in infants with CKD should be evaluated frequently: the total Ca intake from feed, food, and medicines should be within the recommended dietary intake (SDI) in infants with CKD. The Ca requirement may exceed twice the SDI in infants with fast growth or in infants with mineral depleted bone with careful monitoring. The nutritional P intake from feed and food should be within the SDI for age. A phosphate-limited diet must not compromise protein or calcium intake, adding phosphate binders if necessary.

The usage of preterm infant nutritional supplies was suggested as a guide for preterm infants with CKD, correcting their intake according to development and biochemical values. All newborns with CKD, including preterm infants, require taking vitamin D supplements from birth, keeping serum 25(OH)D levels within a target range of 75–120 nmol/L. Active vitamin D in the lowest dose is recommended to reach target PTH and normal calcium concentrations, assuming oral native vitamin D and active vitamin D analogs, avoiding nasogastric or gastrostomy tubes.

Regarding hypocalcemia, oral Ca supplementation in case of insistently high PTH levels is recommended. In cases of acute hypocalcemic emergencies, the use of intravenous Ca is necessary to correct serum Ca levels. Vitamin A supplementation in infants with CKD is not indispensable.

Concerning phosphate supplementation, in infants with CKD it is crucial to preserve P levels within the normal range for age by adjusting diet without compromising protein intake. In cases where serum P is not controlled with optimized nutritional management, it is recommended to introduce P binders; Ca-based P binders are strongly suggested as first-line therapy. In infants with hypercalcemia, it is necessary to consider sevelamer carbonate and, in cases of insistent hyperphosphatemia, to provide P 350 supplementation after optimization of dietary phosphate intake.

In infants undergoing dialysis, it is recommended to increase dialysate Ca concentrations to preserve serum Ca (ionized Ca where existing) concentrations in the normal range. Furthermore, it is suggested to begin the optimization of dialysis in infants with persistently uncontrolled secondary hyperparathyroidism and/or hyperphosphatemia regardless of optimized nutritional and medical management. In infants on dialysis who have persistent and severe hyperparathyroidism in the presence of high or high-normal calcium levels despite optimized conventional management, including active vitamin D, cinacalcet could be considered with carefulness.

Parathyroidectomy may be considered as a last choice treatment when all medical management to limit secondary hyperparathyroidism has failed [44].

### 2.8. Treatment

The prevention and treatment of secondary hyperparathyroidism with a good balance of phosphoremia and calcemia is the principal aim of CKD-CMD therapy [8].

As underlined for infants, the control of serum phosphate with optimized nutritional management is recommended in children. If levels of phosphate cannot be managed, P binders, like Ca-based P binders, need to be considered as a therapeutic option.

As for phosphate, free-calcium binders, such as sevelamer, are currently the first choice for pediatric patients affected by renal failure when dietary regimen is not sufficient; however, calcium carbonate and calcium acetate are also frequently used, with special monitoring of serum calcium concentration [23].

The SDI (suggested dietary intake) for calcium and phosphate in children with CKD2-5D differs depending on age. From 0 to 4 months, the SDI of calcium is 220 mg and the SDI of phosphate is 120 mg; from 4 to 12 months, the SDI of calcium is 330–540 mg and the SDI of phosphate is 275–420 mg; from 1 to 3 years, the SDI of calcium is 450–700 mg and the SDI of phosphate is 250–500 mg; from 4 to 10 years, the SDI of calcium is 700–1000 mg and the SDI of phosphate is 440–800 mg; and from 11 to 17 years, the SDI of calcium is 900–1300 mg and the SDI of phosphate is 640–1250 mg [45].

Regarding intact PTH, the reference values in childhood range from 0.68 pmol/L to 9.39 pmol/L in infants from 6 days to 1 year old; from 1 to 9 years, the range is 1.72 pmol/L to 6.68 pmol/L; from 9 to 17 years, the range is 2.32 pmol/L to 9.28 pmol/L; and from 17 to 19 years, the range is 1.70–6.40 pmol/L [46].

Additionally, magnesium-based compounds are not frequently used use in pediatric CKD because of the risk of hyperkalemia, hypermagnesemia, and diarrhea. Although aluminum-containing agents are active in binding phosphorus, they have been limited because of cumulative bone marrow, skeletal, and central nervous system toxicity [23].

The use of calcimimetics such as cinacalcet in order to reduce PTH secretion is partially limited due to the small amount of available evidence regarding their use in children with chronic renal failure [47,48].

Finally, vitamin D supplementation is really important in this population, as its deficiency contributes to CKD-MBD progression. Recent guidelines suggest administering vitamin D analogues (ergocalciferol or cholecalciferol) in children with CKD 25(OH)D who have persistently increased serum PTH concentrations [49] when serum 25(OH) D levels are below 30 ng/mL [8]. Active vitamin D analogues (calcitriol, alfacalcidol, paricalcitol, or doxercalciferol) should be used in cases of serum PTH above target range and serum 25(OH)D over 30 ng/mL with contextual absence of hypercalcemia and hyperphosphatemia [8]. Daily oral calcitriol is secure and well tolerated by these patients, with a low incidence of consequent hypercalcemia [49]. Different authors agree that vitamin D analogues should be administered in the lowest dose to achieve target PTH concentrations and maintain normal serum calcium concentration, performing monthly controls of blood PTH, calcium and phosphate in the first three months and every three months thereafter [49]. Clinicians should start with an intensive replacement phase of three months and a subsequent maintenance phase. The treatment should be discontinued when 25(OH) D concentration is over 48 ng/mL or if the child is hypercalcemic [8,49].

A lot of data suggest that regular and early implementation of both aerobic and resistance exercise programs in children and adolescent with chronic kidney disease have positive effects on muscle function, exercise tolerance, and quality of life. Physical activity is highly encouraged in CKD patients; resistance exercise promotes muscle mass and strength, while endurance training enhances physical performance and exercise capacity [14].

Metabolic acidosis is associated with poor growth in children with CKD. Treatment of metabolic acidosis may lead to complications such as volume overload, exacerbation of hypertension, and facilitation of vascular calcifications. However, several adverse consequences have been associated with metabolic acidosis, including impaired albumin biosynthesis and glucose metabolism, increased protein catabolism with a decrease in muscle mass, and increased bone resorption and inhibition of its formation with growth retardation in children. Giving bases aimed at normalization of plasma bicarbonate concentration might be necessary in some cases. At present, the KDIGO recommendations support maintaining HCO3− concentration ≥ 22 mmol/L in all CKD patients using oral sodium bicarbonate and/or the use of HCO3− based or lactate-based dialysis solutions in patients on dialysis [50,51]

Growth hormone therapy is recommended in children with stage 3–5 CKD or on dialysis if they have persistent growth failure, defined as a height below the third percentile for age and sex and a height velocity below the twenty-fifth percentile, once other potentially treatable risk factors for growth failure have been adequately evaluated and it can be stated that the child has growth potential. In children who have received a kidney transplant and fulfil the above growth criteria, initiation of GH therapy is recommended 1 year after transplantation if spontaneous catch-up growth does not appear and steroid-free immunosuppression is not a feasible option. GH should be given at dosages of 0.045–0.05 mg/kg per day by daily subcutaneous injections until the patient has reached their final height or until renal transplantation [13].

## 3. Conclusions and Future Perspectives

Pediatric CKD is a vigorous and multifaceted disease with exclusive aspects that diverge this population from adults. Due to rapid bone growth and efforts with nutrition, the acceptable control of CKD-MBD in infants is especially challenging.

Pediatric nephrology research groups have performed estimable and exhilarating efforts concentrating on this group, and evidence-based management of this population is growing, but for now there are still very few high-quality findings to guide evidence-based practice.

In the future, it may be necessary to define the targets for the principal biomarkers in this age group in case of CKD, especially PTH, 25(OH)D, and ALP. Studies to establish the true calcium necessities in infants for optimal bone quality and without a risk of vascular calcifications are needed. Evaluating the incidence and risk factors for fractures in children with CKD and on dialysis is crucial. Moreover, it would be important to detect the required quantity of Ca intake and the best PTH target range, which allow for normal turnover and mineralization (determined by histology, imaging, and biomarker studies) without deteriorating vascular calcifications. More data are necessary to establish if urinary calcium excretion in healthy infants and those with CKD can be used to estimate optimal calcium intakes. Evaluation of the importance of cortical bone assessment in bone biopsy cores to predict fracture risk and evaluation of the sensitivity and specificity of DXA to predict fracture risk in pediatric CKD would be crucial in the future. Further investigations are also required to find novel CKD-MBD serum biomarkers that would allow for a more precise indirect evaluation of bone histomorphometry, cell maturation, and skeletal mineralization. At the same time, innovative techniques should be addressed toward the assessment of osseous qualitative and biomechanical properties, such as microarchitecture, accumulated microscopic damage, the quality of collagen, and the size of mineral crystals.

Finally, our understanding of the link between mineral disturbances and vascular calcification in CKD requires further study. An important issue for future research is to find a balance of calcium and phosphate intake in young patients in order to optimize skeletal mineralization while reducing the risk of vascular calcification due to excessive calcium intake. An accurate estimation of the real-time changes in bone mineral balance may guide treatments based on the individual’s state of bone turnover and mineralization.

## Figures and Tables

**Table 1 biology-12-01395-t001:** Suggested interval of clinical, biochemical, and radiological assessments (in months) for CKD-MBD in children by CKD stage and age.

Age	Assessments	Mild CKD	Moderate CKD	Severe CKD and Dialysis
0–1 year	Clinical evaluation	1–3	0.5–2	0.25–1
	Biochemicals (Ca, P, HCO3−, PTH, ALP)	3–6	1–3	0.25–1
	Biochemicals (25OH)	6	3–6	3
	X-rays	Only if clinical signs	Only if clinical signs	Only if clinical signs
1–3 year	Clinical evaluation	3–6	1–3	0.5–2
	Biochemicals (Ca, P, HCO3−, PTH, ALP)	3–6	1–3	0.5–1
	Biochemicals (25OH)	6–12	3–6	3
	X-rays	Only if clinical signs	Only if clinical signs	Only if clinical signs
>3 year	Clinical evaluation	3–6	1–3	1–3
	Biochemicals (Ca, P, HCO3−, PTH, ALP)	6	3–6	1–3
	Biochemicals (25OH)	6–12	3–6	3
	X-rays	Only if clinical signs	Only if clinical signs	Only if clinical signs
Puberty	Clinical evaluation	3–6	1–3	1–3
	Biochemicals (Ca, P, HCO3−, PTH, ALP)	6	3–6	1–3
	Biochemicals (25OH)	6–12	3–6	3
	X-rays	Only if clinical signs	Only if clinical signs	Only if clinical signs

**Table 2 biology-12-01395-t002:** Comparing bone imaging techniques in children with CKD-MBD.

	Plain Radiography	DXA	QCT, pQCT, and HR-pQCT	MRI	US
Main evaluated parameters	Gross evaluation of mineralization	Bone mineral density and body composition	Bone microarchitecture, biomechanics and volumetric mineralization; prediction of fracture risk	Bone microarchitecture, volumetric mineralization and soft tissue evaluation (bone muscle unit)	Cortical bone evaluation
Skeletal site	All skeleton	Hip, distal radius, lumbar spine, all skeleton	Hip, distal radius, distal tibia	Distal radius, distal tibia, calcaneus, hip, spine, all skeleton	Tibia
Availability	++++	++++	++	++	+++
Cost	+	+	++	+++	+
Radiation exposure	Yes	Yes, minor	Yes, minor	No	No
Presence of reference data	Present	Present	Present	Absent	Absent
Main concerns	Observer dependent interpretation, bidimensionality, low sensitivity	Bidimensionality, underestimation of mineralization, low resolution, difficult follow up of growing bones	No consensus regarding the drawing of the reference line and main regions of interest	No consensus regarding the main regions of interest	Operator dependent

+: normal; ++: elevated; +++: moderately elevated; ++++: severely elevated.

## Data Availability

Not applicable.

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
