# Peer review of "Bone Disorders in Pediatric Chronic Kidney Disease: A Literature Review"

_biology, 2023, doi:10.3390/biology12111395_

Round 1

Reviewer 1 Report

Comments and Suggestions for Authors

This is a comprehensive summary about the bone disorders in children with chronic kidney disease.  

1.My main concern is that by applying this research methodology, it is abstract how the inappropriate papers were eliminated.  

2. In the Pathogenesis section, from lines 124 to 151 there are no references at all. More specifically: In lines 143-146 the authors point out the possible effect of muscle on bone development. Please add the appropriate reference for CKD pediatric population. In lines 147-149 the authors mention about the term osterosarcopenia in pediatric CKD. Are there any studies performed in this issue in pediatric population? Moreover they mention that muscle atrophy and weakness is present in 7-20% of cases. Please add a reference to this sentence. In lines 149-150, the authors state “Again, vitamin D deficiency and elevated PTH levels have a negative effect on muscle.” How do they base this sentence? Are there any relevant studies performed in pediatric population? In line 152 the authors remark the effect of inflammation on myostatin expression, but they do not mention the pediatric studies regarding the effect of inflammation on mineral bone parameters. Moreover, the authors do not describe the effect of anemia and iron balance on mineral bone disorders. 

3. In the section Assessing bone mineralization in line 222 the authors mention the need for monitoring serum bicarbonate. Why? Please explain. 

4. In line 283 the authors state that "Pilot studies suggest that vitamin D therapy for secondary hyperparathyroidism may confer a cardiac protection and decrease mortality". They need to include a reference to this sentence. 

6. In lines 286-290 the authors describe the metabolic syndrome and the obesity in pediatric CKD patients. Where is the link between this condition and mineral bone disorders? 

7. In the treatment section, there is no mention about the effect of correction of metabolic acidosis, GH therapy, physical exercise on bone health and possible novel treatments for the management of bone disorders in pediatric CKD. 

Reviewer 2 Report

Comments and Suggestions for Authors

Excellent reviewed paper. The nephrology literature is full of papers on CKD-MBD in adults. Even the guidelines are mostly directed at adults. The problem of CKD MBD in children is less often written about.

This reviewed article will surely attract the attention of everyone who is interested in mineral metabolism disorders and bone diseases.

Reviewer 3 Report

Comments and Suggestions for Authors

The authors attempted a literature review of pediatric CKD-MBD, examining the etiology, diagnostic approaches, and treatment of CKD-MBD, as well as the impact of renal osteodystrophy on the growing skeleton. As the authors describe, studies investigating the management of CKD-MBD in children from a biological basis are limited, making this review interesting, but also raising the following concerns:

1. Method: The search expressions used by the authors in their literature search are ambiguous. The search expressions used by the authors in their literature search are ambiguous. For example, using "Renal AND Bone" as the search expression, PubMed found over 35,000 references. How did the authors extract appropriate studies from them? This should be done more systematically.

2. The authors state in the introduction that this review focuses specifically on the biological basis of the pathogenesis of pediatric CKD-MBD, but most of the content is a general description of clinical aspects of CKD-MBD or not limited to pediatric patients. Couldn't the authors find any literature that studied the biological basis of the pathophysiology more characteristic of pediatric CKD-MBD?

3. I understand that CKD under 2 years of age is difficult to manage, but what is the reason for having a separate section for CKD-MBD for those under 2 years of age? The problem of hyperphosphatemia should be the same for infants over 2 years of age. I think management of hyperphosphatemia should be in the "treatment" section. If CKD-MBD under 2 years of age has different biological features than children over 2 years of age, the authors should highlight them here.

4. Treatment:

In the management of CKD-MBD, reference values for laboratory measurements should be different depending on age, and treatment targets should be different as well. In particular, intact-PTH may differ depending on the CKD stage, but how about at least providing age-specific reference values and targets for phosphorus and iPTH?

The authors should touch on recombinant growth hormone therapy as management of CKD-MBD in children.

Round 2

Reviewer 1 Report

Comments and Suggestions for Authors

The authors have adequately responded to my comments. I have no further comments.

Reviewer 3 Report

Comments and Suggestions for Authors

This revised manuscript has been significantly improved by appropriately addressing the reviewers' comments. I have no further comments or questions.